# Do Elderly Patients with Atrial Fibrillation Have Comparable Ablation Outcomes Compared to Younger Ones? Evidence from Pooled Clinical Studies

**DOI:** 10.3390/jcm11154468

**Published:** 2022-07-31

**Authors:** Feng Li, Lei Zhang, Li-Da Wu, Zhi-Yuan Zhang, Huan-Huan Liu, Zhen-Ye Zhang, Jie Zhang, Ling-Ling Qian, Ru-Xing Wang

**Affiliations:** Department of Cardiology, Wuxi People’s Hospital Affiliated to Nanjing Medical University, Wuxi 214023, China; lifeng212910@njmu.edu.cn (F.L.); leizhanglz@163.com (L.Z.); lidawunjmu@outlook.com (L.-D.W.); zzy18862935193@163.com (Z.-Y.Z.); hhliu1224@163.com (H.-H.L.); zhangzhenye@njmu.edu.cn (Z.-Y.Z.); jiezhang980216@163.com (J.Z.); qll900@sina.com (L.-L.Q.)

**Keywords:** ablation, atrial fibrillation, elderly patients, younger patients, outcomes

## Abstract

**Background:** Age is an independent risk factor of the progress and prognosis of atrial fibrillation (AF). However, ablation outcomes between elderly and younger patients with AF remain elusive. **Methods:** Cochrane Library, Embase, PubMed, and Web of Science were systematically searched up to 1 April 2022. Studies comparing AF ablation outcomes between elderly and younger patients and comprising outcomes of AF ablation for elderly patients were included. Trial sequential analysis (TSA) was performed to adjust for random error and lower statistical power in our meta-analysis. Subgroup analysis identified possible determinants of outcome impact for elderly patients after ablation. Moreover, linear and quadratic prediction fit plots with confidence intervals were performed, as appropriate. **Results:** A total of 27 studies with 113,106 AF patients were eligible. Compared with the younger group, the elderly group was significantly associated with a lower rate of freedom from AF (risk ratio [RR], 0.95; *p* = 0.008), as well as a higher incidence of safety outcomes (cerebrovascular events: RR, 1.64; *p* = 0.000; serious hemorrhage complications: RR, 1.50; *p* = 0.035; all-cause death: RR, 2.61; *p* = 0.003). Subgroup analysis and quadratic prediction fit analysis revealed the follow-up time was the potential determinant of freedom from AF for elderly patients after AF ablation. **Conclusions:** Our meta-analysis suggests that elderly patients may have inferior efficacy and safety outcomes to younger patients with AF ablation. Moreover, the follow-up time may be a potential determinant of outcome impact on freedom from AF for elderly patients after AF ablation.

## 1. Introduction

Atrial fibrillation (AF) has become the most common sustained cardiac arrhythmia worldwide, with an estimated prevalence ranging from 2% to 4% in adults. Approximately 12 million individuals will experience AF in the US by 2050 and nearly 18 million in Europe by 2060. Remarkably, the prevalence could increase to as high as 5–10% among those aged 65 years and older [1,2]. Meanwhile, accumulated studies have reported that the elderly population with AF has a high risk of arrhythmic burden, stroke, bleeding, and heart failure, ultimately leading to longer hospitalization and increased mortality. Additionally, old age is an independent risk factor of AF progression and prognosis [3]. Therefore, the prevention and management of AF in elderly individuals have been hotpots in the cardiac electrophysiology field.

Ablation has been demonstrated as an effective strategy for rhythm control and life-quality improvement in symptomatic and drug-refractory AF patients [4,5]. The latest AF guidelines emphasized that catheter ablation for selected elderly AF patients might be a safe and effective option with comparable success rates and acceptable complication incidence in younger AF patients [3]. However, recommendations to date have not been made in either the European Society of Cardiology or American Heart Association guidelines, which indicates a potentially unresolved controversy in AF ablation therapy for the elderly. Alternatively, a recent retrospective study on the efficacy of second-generation cryoballoon ablation for elderly patients with persistent AF reported that old individuals (≥75 years) were significantly associated with a higher AF recurrence than younger individuals (63.9 vs. 53.0%, *p* = 0.03) with the median follow-up of 24 months [6].

The results comparing ablation outcomes between elderly and younger patients with AF remain elusive and are thus vigorously debated. Accordingly, we aimed to perform a meta-analysis of a relatively large sample to comprehensively evaluate ablation outcomes between elderly and younger patients with AF.

## 2. Methods

### 2.1. Study Design

This systematic review was performed based on the PRISMA guidelines. The registered protocol for our study is available in the PROSPERO database (https://www.crd.york.ac.uk/prospero/display_record.php?ID=CRD42022325471, accessed on 10 May 2022).

### 2.2. Search Strategy

Two independent reviewers (F. Li and L. Zhang) comprehensively searched four online databases—the Cochrane Library, Embase, PubMed, and Web of Science—from their establishment to 1 April 2022. Search keywords were “elderly”, “older”, “septuagenarians”, “octogenarians”, “nonagenarians”, “centenarians”, “younger”, “atrial fibrillation”, and “ablation”. Trials on the outcomes of AF ablation for elderly patients or comparing ablation outcomes between elderly patients and younger patients with AF were included. A manual search of reference lists of review literature and retrieved eligible literature was performed for potential publications not identified previously. In addition, we also contacted the relevant corresponding authors for missing outcome data in their publications.

### 2.3. Search Design

Two independent reviewers (L.-D. Wu and Z.-Y. Zhang) searched and reviewed the titles, abstracts, and full texts to select the eligible studies. A study was eligible if meeting the following inclusion criteria: randomized controlled trials and cohort, observational studies, and single-arm studies; studies comparing AF ablation outcomes between elderly patients and younger patients, including efficacy outcome (e.g., freedom from AF) and safety outcomes (e.g., cerebrovascular events, serious hemorrhage complications, phrenic nerve injury, and all-cause death); studies on outcomes of AF ablation for elderly patients; and studies with full text published in peer-reviewed journals; and studies containing the most data for multiple publications of the same study. Studies without original data, case reports, editorials, review articles, letters, and animal studies were all excluded. Meanwhile, a third reviewer (R.-X. Wang) resolved any disagreements about eligibility.

### 2.4. Data Extraction and Quality Assessment

Two independent researchers (F. Li and L. Zhang) extracted the data for each eligible study, and any disagreements were settled by a third researcher (R.-X. Wang). First, we documented study characteristics: first author, publication year, study design, country, sample size in the elderly and younger groups, and follow-up time. Then, patients’ demographic and clinical characteristics and procedure-related indices were also recorded.

Study quality was evaluated using two appraisal tools by two independent researchers (L. Zhang and L.-D. Wu). For two-arm observational studies, the Newcastle–Ottawa Quality Assessment Scale was used, which has three domains with nine points [7]. The quality levels of studies were divided into moderate-to-high quality (score ≥ 6) and low quality (score < 6). For the single-arm study, the Institute of Health Economics checklist with a total of 20 items was used, and scores ranged from 0 (poor) to 20 (excellent) [8]. Any disagreements were discussed and resolved by consulting a third researcher (R.-X. Wang).

### 2.5. Statistical Analysis

Continuous variables are displayed as means ± standard deviations or medians with interquartile ranges, and categorical variables are displayed as frequencies and percentages. Relative risk (RR) and corresponding 95% confidence intervals (CIs) were calculated for each outcome for observational studies with two arms, whereas in terms of single-arm analysis, pooled results are presented as incidence of the events (event numbers divided by patient numbers) and 95% CI. Stata version 12.0 (http://www.stata.com, accessed on 10 May 2022) was used for statistical analyses. *p* < 0.05 was considered statistically significant.

We used the chi-squared test and I-squared (I^2^) to quantify and assess statistical heterogeneity among studies. If the I^2^ value was more than 50% and/or *p* < 0.05 for the chi-squared test, we considered the between-study heterogeneity to be substantial, and a random-effect model was used. Otherwise, a fixed-effect model was used. Sensitivity analysis was performed to assess the effect of a single study on the overall risk by sequentially omitting one study at a time, and potential publication bias was also evaluated via Egger’s and Begg’s tests. Importantly, trial sequential analysis (TSA), a useful method providing the required information size (RIS), was performed by TSA viewer (version 0.9.5.10 beta; Copenhagen Trial Unit) to adjust the random error and lower statistical power caused by the limited number of trials in a meta-analysis [9]. Type I and type II errors were set to 5% and 20% (80% power), respectively.

In addition, subgroup analysis was performed to screen sources of heterogeneity and potential determinants of AF ablation outcomes between elderly and younger patients. According to the characteristics of eligible studies, some potential factors, and previously reported factors, 13 subgroup factors were identified: study design, publication date, elderly age cutoff, elderly group sample size, AF type, female proportion, hypertension proportion, diabetes mellitus (DM) proportion, LAD (left atrial diameter), AF history duration, ablation strategy, ablation energy, and follow-up time. If the study design included only one center, it was defined as a single-center subgroup; otherwise, it was defined as a multicenter subgroup. If the publication date was within nearly 3 years (2020–2022), it was defined as the “recently” subgroup; otherwise, it was defined as the “not recently” subgroup. According to cutoff values of 75 and 100, the elderly age cutoff and the elderly group sample size were divided into two subgroups, respectively. If the AF types of the eligible patients were all paroxysmal AF (PAF) in the elderly group, it was assigned to the PAF subgroup; otherwise, it was assigned to the non-PAF subgroup. If the female proportion was significantly higher (elderly group versus younger group), it was defined as a “higher” subgroup; if the female proportion was equal, it was defined as an “equal” subgroup. Similarly, hypertension, DM proportion, LAD, and AF history duration with “higher” and “equal” subgroup also were defined, respectively. If the ablation strategy included only PVI, it was assigned to the PVI subgroup, and if PVI plus linear and/or substrate ablation had been performed, it was assigned to the “PVI-plus” subgroup. Based on the energy source of radiofrequency ablation (RF) and cryoablation (Cryo), RF and Cryo subgroups were defined. Follow-up time was divided into two subgroups (≥24 months and <24 months).

Additionally, linear and quadratic prediction fit plots with confidence intervals were constructed as appropriate to assess the correlation between the follow-up time and the rate of freedom from AF for elderly and younger patients.

## 3. Results

### 3.1. Study Selection and Quality Assessment

A total of 27 studies with 113,106 AF patients (8686 elderly patients and 104,420 younger patients) were eligible, including 23 observational two-arm studies [6,10,11,12,13,14,15,16,17,18,19,20,21,22,23,24,25,26,27,28,29,30,31] (8133 elderly AF patients and 104,420 younger AF patients) and four single-arm studies [32,33,34,35] (553 elderly AF patients). The selection flowchart is displayed in Figure 1. Two studies (Romero et al. [18] and Hao et al. [28]) reported only the safety outcomes without the rate of freedom from AF. A total of five studies contained multiple age-based subgroups: four subgroups in Sciarra et al. [11], six subgroups in Hartl et al. [17], five subgroups in Bunch et al. [25], three subgroups in Kusumoto et al. [30], and three subgroups in Bhargava et al. [31]. Therefore, for these five studies, multiple subgroups were integrated into two groups (elderly group and younger group) based on the elderly age cutoff value in each study. The baseline characteristics and procedure-related indices of the eligible studies are presented in Table 1. In this meta-analysis, all two-arm studies had a moderate-to-high quality, as presented in Appendix A. Four single-arm studies all had a score higher than 15, and are presented in Appendix A.

### 3.2. Rate of Freedom from AF between Elderly and Younger Groups

A total of 21 studies [6,10,11,12,13,14,15,16,17,19,20,21,22,23,24,25,26,27,29,30,31] on 18,608 AF patients in our meta-analysis reported rates of freedom from AF between elderly and younger groups. Compared with the younger group, the elderly group was significantly associated with a lower rate of freedom from AF (RR, 0.95; 95% CI, 0.92–0.99; *p* = 0.008; I^2^ = 46.30%; Figure 2) via a random-effect model.

Subgroup analysis was performed with a total of 11 subgroup factors for the rate of freedom from AF, and the results are displayed in Figure 3. Elderly group sample size subgroup analysis showed a comparable rate of freedom from AF between the elderly group and the younger group: ≥100 subgroup (RR 0.96; 95% CI, 0.93–1.00; *p* = 0.054) and <100 subgroup (RR 0.92; 95% CI, 0.85–1.00; *p* = 0.050). Similar results were also shown in the LAD and AF history duration subgroups (Appendix A). Compared with the younger group, the elderly group was significantly associated with a lower rate of freedom from AF in the single-center subgroup (RR 0.94; 95% CI, 0.89–0.99; *p* = 0.014), recently subgroup (RR 0.90; 95% CI, 0.80–1.00; *p* = 0.041), <75 years subgroup (RR 0.93; 95% CI, 0.88–0.98; *p* = 0.006), non-PAF subgroup (RR 0.95; 95% CI, 0.91–0.99; *p* = 0.012), higher female proportion subgroup (RR 0.94; 95% CI, 0.89–0.99; *p* = 0.022), higher hypertension proportion subgroup (RR 0.95; 95% CI, 0.91–1.00; *p* = 0.037), higher DM proportion subgroup (RR 0.95; 95% CI, 0.91–0.99; *p* = 0.010), PVI-plus subgroup (RR 0.93; 95% CI, 0.89–0.98; *p* = 0.010), and RF subgroup (RR 0.95; 95% CI, 0.91–0.99; *p* = 0.009), all of which were consistent with the pooled results, whereas no significant differences were found in the other subgroups. Importantly, the only potentially significant treatment–covariate interaction was identified in the follow-up time subgroups: ≥24 months (RR 0.87; 95% CI, 0.78–0.97; *p* = 0.015) and <24 months (RR 0.97; 95% CI, 0.94–1.00; *p* = 0.075) with *p* = 0.066 for interaction.

Sensitivity analysis was also performed, and the results showed no significant change, ranging from 0.94 (95% CI, 0.90–0.99) to 0.96 (95% CI, 0.93–1.00) in the overall combined proportion, which suggested that no single study dominated the combined proportion and heterogeneity. Moreover, no publication bias was presented in Begg’s or Egger’s test (*p* = 0.174 and *p* = 0.115, respectively).

In addition, TSA was used to assess whether there was adequate power for comparison of rates of freedom from AF between the elderly group and the younger group. The results showed that although the actual sample size (18,608) was smaller than the RIS (relative risk reduction, RRR = 35%; RIS = 25,240), the cumulative Z curve (Z = 2.55) crossed both the conventional boundary and the trial sequential alpha spending monitoring boundary (TSA monitoring boundaries = 2.49), suggesting firm evidence favoring the younger group in terms of the rate of freedom from AF (Figure 4).

### 3.3. Pooled Rate of Freedom from AF in Elderly Group

A total of 25 eligible studies (3879 elderly patients with AF) reported the rate of freedom from AF [6,10,11,12,13,14,15,16,17,19,20,21,22,23,24,25,26,27,29,30,31,32,33,34,35]. The pooled rate of freedom from AF was 0.69 (95% CI, 0.63–0.75; *p* = 0.000; I^2^ = 92.53%; Figure 5) with the random-effect model.

Subgroup analysis was performed with a total of eight subgroup factors for the rate of freedom from AF in the elderly group, and the results are shown in Table 2. Interestingly, in terms of follow-up time for elderly patients, a significantly lower rate of freedom from AF was shown in the ≥24 months subgroup (0.53; 95% CI, 0.43–0.62; *p* = 0.000; I^2^ = 85.67%) than in the <24 months subgroup (0.76; 95% CI, 0.71–0.81; *p* = 0.000; I^2^ = 86.11%) with *p* = 0.000 for interaction.

Sensitivity analysis indicated that there was no significant change, ranging from 0.68 (95% CI, 0.62–0.74) to 0.70 (95% CI, 0.65–0.75), in the overall combined proportion, indicating that no single study dominated combined proportion and heterogeneity. No publication bias was shown in Begg’s or Egger’s test (*p* = 0.350 and *p =* 0.277, respectively). Therefore, the results were considered to be robust.

### 3.4. Relationship between Follow-Up Time and Rate of Freedom from AF

In terms of the elderly group, linear prediction fit and quadratic prediction fit plots with confidence intervals were constructed. The results showed that the correlation between the follow-up time and the rate of freedom from AF was significantly negative (quadratic prediction fit: R^2^ = 57.10%, *p* = 0.000; linear prediction fit: R^2^ = 55.90%, *p* = 0.000) (Figure 6). In addition, overlapping of the quadratic prediction fit plots with confidence intervals for elderly and younger groups was performed, and the result indicated that the rate of freedom from AF in the elderly group seemed to be consistently lower than in the younger group (Appendix A). More interestingly, we took the quadratic prediction fit curve for the younger group minus the quadratic prediction fit curve for the elderly group, which calculated the difference in freedom from AF between the younger group and the elderly group. The result showed that the difference was monotonically decreased in the interval of 0 to 20.98 months, while the difference was monotonically increased in the interval of ≥20.98 months (Appendix A).

### 3.5. Safety Outcomes between Elderly and Younger Groups

A total of 21 eligible studies [6,11,12,13,14,15,16,17,18,19,20,21,22,23,24,25,26,27,28,29,31] compared cerebrovascular events, including stoke or transient ischemic attack (TIA), between elderly and younger groups, whereas a total of 6 studies [6,14,15,19,23,27] were excluded because there were no events in either group. The result showed that the elderly group had a significantly higher incidence of cerebrovascular events than the younger group (RR, 1.64; 95% CI, 1.25–2.17; *p* = 0.000; I^2^ = 0.00%) with a fixed-effect model (Figure 7). However, the pooled rate of the cerebrovascular events [6,11,12,13,14,15,16,17,18,19,20,21,22,23,24,25,26,27,28,29,31,32,33,35] in the elderly group was 0.00 (95% CI, 0.00–0.01; *p* = 0.000) (Appendix A).

A total of 19 eligible studies [6,11,12,13,14,15,16,17,19,20,21,22,23,26,27,28,29,30,31] reported serious hemorrhage complications (such as hemothorax, perforation, tamponade, or major bleeding) between the elderly and younger groups, whereas two studies [14,23] were excluded because of no events in either group. The result indicated that the elderly group had a significantly higher incidence of serious hemorrhage complications (RR, 1.50; 95% CI, 1.03–2.19; *p* = 0.035; I^2^ = 0.00%) with a fixed-effect model (Figure 8). However, the pooled rate of serious hemorrhage complications [6,11,12,13,14,15,16,17,19,20,21,22,23,26,27,28,29,30,31,32,33,35] in the elderly group was 0.00 (95% CI, 0.00–0.01; *p* = 0.000) (Appendix A).

All-cause death was reported by 14 eligible studies [6,12,13,14,16,17,18,19,22,23,24,25,27,29] between elderly and younger groups. A total of five studies [6,14,19,23,27] without events in either group were excluded from the meta-analysis. The results also showed that the elderly group had a significantly higher incidence of all-cause death (RR, 2.61; 95% CI, 1.38–4.93; *p* = 0.003; I^2^ = 65.80%) with a random-effect model (Figure 9). The pooled rate of all-cause death [6,12,13,14,16,17,18,19,22,23,24,25,27,29,32,33] in the elderly group was 0.01 (95% CI, 0.00–0.02; *p* = 0.050) (Appendix A).

In terms of phrenic nerve injury, 10 [6,11,12,16,17,19,21,22,23,28] of 12 eligible studies [6,11,12,13,16,17,19,21,22,23,28,29] were analyzed in our meta-analysis. The result showed there was no significant difference between elderly and younger groups (RR, 0.90; 95% CI, 0.62–1.31; *p* = 0.587; I^2^ = 0.00%) with a fixed-effect model (Figure 10). The pooled rate of phrenic nerve injury [6,11,12,13,16,17,19,20,21,22,23,28,29,32,33] in the elderly group was 0.01 (95% CI, 0.00–0.02; *p* = 0.011) (Appendix A).

## 4. Discussion

We comprehensively evaluated a total of 113,106 AF patients (8686 elderly patients and 104,420 younger patients) from 27 original articles. To our knowledge, this study may be the first registered meta-analysis with a relatively large sample to compare the AF ablation outcomes between elderly patients and younger patients. Our main findings were as follows. (1) Elderly patients may have inferior efficacy and safety outcomes to younger patients with AF ablation. (2) Follow-up time may be a potential determinant of outcome impact on freedom from AF for elderly patients after AF ablation.

The prevalence of AF has been reported to increase with the aging progress, which leads to age as an independent risk factor of the progress and prognosis of AF [2,36]. At present, the diagnosis and management of AF guidelines recommend that catheter ablation should be considered in PAF and persistent AF patients for better symptom control (Class I) [3]. A prespecified age-subgroup analysis based on the CABANA trial also reported that AF recurrence rates were consistently lower with ablation than with drug therapy across three age subgroups, with adjusted hazard ratio (aHR) of 0.47 (95% CI, 0.35–0.62) for <65 years, aHR 0.58 (95% CI, 0.48–0.70) for 65–74 years, and aHR 0.49 (95% CI, 0.34–0.70) for ≥75 years, with *p* = 0.396 for interaction [37], indicating support for ablation without justification to discriminate by age. Several observational studies revealed a possible trend wherein elderly patients might be associated with a lower rate of freedom from AF and a higher rate of safety outcomes than younger patients [6,38], which meant that ablation outcomes between elderly and younger AF patients remain poorly understood. Therefore, a systematic review is needed to pool existing data and assess the AF ablation outcomes between elderly and younger patients.

In this meta-analysis, we compared the ablation outcomes between elderly and younger AF patients. For the rate of freedom from AF, the results indicated a pooled rate of freedom from AF in elderly patients of 0.69 (95% CI, 0.63–0.75; *p* = 0.000) and a significantly lower rate of freedom from AF in elderly patients than younger patients (RR, 0.95; *p* = 0.008), which was consistent with previous studies [6,32,33,38]. Moreover, the TSA result also showed firm evidence favoring the younger group in terms of the rate of freedom from AF, which further supported our pooled results. In terms of safety outcomes, we found a relatively low incidence (ranging from 0.00 to 0.01) of cerebrovascular events, serious hemorrhage complications, phrenic nerve injury, and all-cause death for elderly AF patients, whereas compared with younger ones, a significantly higher rate of cerebrovascular events, serious hemorrhage complications, and all-cause death was shown in elderly patients with AF, while a comparable rate of phrenic nerve injury was displayed in elderly patients. In summary, elderly patients may have inferior efficacy and safety outcomes to younger patients with AF ablation.

Previous studies revealed that the success rate of maintenance of sinus rhythm was progressively reduced as a function of follow-up time postablation, ranging from 75% to 93% and 63% to 74% for PAF and persistent AF, respectively, with 1-year follow-up, as well as ranging from 57% to 65% and <50% for PAF and persistent AF, respectively, with 5-year follow-up [39,40]. In our study, we found the rate of freedom from AF between elderly and younger groups in the ≥24 months subgroup (RR 0.87; *p* = 0.015) was lower in the <24 months subgroup (RR 0.97; *p* = 0.075) with *p* = 0.066 for interaction, which indicated a substantial potential trend between ≥24 months follow-up and <24 months follow-up. Meanwhile, a significantly lower rate of freedom from AF was shown in the ≥24 months subgroup (pooled rate: 0.53; *p* = 0.000) than in the <24 months subgroup (pooled rate: 0.76; *p* = 0.000) with *p* = 0.000 for interaction in terms of follow-up time for elderly patients, suggesting that extended follow-up time may be severely detrimental to freedom from AF for elderly patients after ablation. Similarly to previous studies [41,42,43], our quadratic prediction fit result for the elderly group showed that the correlation between the follow-up time and the rate of freedom from AF was significantly negative. In addition, we analyzed the difference in rates of freedom from AF between younger and elderly groups by means of the quadratic prediction fit curves. The result showed the rate of freedom from AF in the younger group seemed to be consistently higher than in the elderly group, as well as the difference between the rates of freedom from AF was increasing when the follow-up time was more than 20.98 months, which might provide a promising explanation for the potential trend in the follow-up time subgroup between elderly and younger groups.

Multiple risk factors, including unmodifiable risk factors (e.g., gender, age, and genetics) and modifiable risk factors (e.g., hypertension, DM, and obesity), played a significant role in contributing to the initiation and progression of AF. A growing number of clinical studies have suggested that female patients have a higher risk of AF recurrence than male patients after ablation, owing to more advanced atrial remodeling [44,45]. The recurrent AF subanalysis in the CABANA trial (NCT00911508) proved that the efficacy of ablation in the female subgroup was significantly inferior to that in the male subgroup when compared with the efficacy of drug therapy (HR: 0.64 vs. 0.46, *p* for interaction = 0.035) [46]. Similarly, compared with the younger group, the elderly group was significantly associated with a lower rate of freedom from AF in the higher female proportion subgroup, while a comparable rate of freedom from AF was found in the equal female proportion subgroup, which in part suggested that the higher the proportion of females, the lower the rate of freedom from AF. In addition, the latest subanalysis from the CABANA trial indicated that a superior result for reducing AF recurrence was displayed in the catheter ablation arm than in the drug therapy arm across age-groups [37]. Interestingly, our results showed that compared with the younger group, the elderly group was significantly associated with a lower rate of freedom from AF with the elderly age cutoff less than 75 years, while a comparable freedom rate from AF when the elderly age cutoff was more than 75 years. The reason might be the lower the elderly age cutoff and the higher the proportion of young patients in the younger group, ultimately contributing to a better prognosis in the younger group. Moreover, LAD, AF history duration, and AADs usage were also reported to affect the rate of freedom from AF ablation, whereas our results showed that subgroup analysis in terms of the LAD and AF history duration subgroups showed no significant difference between the equal and higher subgroups (*p* = 0.168 and *p* = 0.685, respectively), which might be attributed to the relatively small absolute difference between the two groups in the values of LAD and AF history duration, as well as there being fewer studies with higher LAD and AF history duration between elderly and younger groups. In addition, AADs usage was similar between the elderly and younger groups in most of the eligible studies.

Reportedly, hypertension and DM are the two most common cardiovascular risk factors and comorbidities, with a higher risk of developing AF and a lower rate of freedom from AF postablation than normotensives and non-DM patients, respectively [3,47]. Interestingly, the elderly group was significantly associated with a lower rate of freedom from AF in the higher hypertension proportion subgroup and the higher DM proportion subgroup, while comparable freedom rates from AF were found in the equal hypertension proportion subgroup and the equal DM proportion subgroup, respectively. A significant clinical implication underlying this result was that good management of hypertension and DM might be expected to extensively improve ablation efficacy for elderly patients with AF.

Additionally, our results indicated that compared with the younger group, the elderly group was significantly associated with a lower rate of freedom from AF in the single-center subgroup, recently subgroup, non-PAF subgroup, PVI-plus subgroup, and RF subgroup, while a comparable freedom rate from AF were found in the multicenter subgroup, not-recently subgroup, PAF subgroup, PVI subgroup, and Cryo subgroup, which indicated elderly patients might have benefits similar to younger patients in these subgroups. These results must still be demonstrated with more randomized controlled trials, which might play a guiding role on the optimal management of AF for elderly individuals.

## 5. Limitations

Several limitations in this meta-analysis should be highlighted. First, a major limitation is that all the eligible studies were observational studies without randomized controlled trials, which might restrict us from drawing a substantial conclusion. To the best of our knowledge, this study was the first registered meta-analysis with a relatively large sample to compare AF ablation outcomes between elderly and younger patients. The TSA result also indicated firm evidence supporting our pooled results. Second, consistently with previous meta-analyses, possible biases might have affected our results. In this study, sensitivity analysis and publication bias tests (e.g., Begg’s and Egger’s tests) both indicated that our results were robust. Third, similar to previous studies [18,25], our safety results showed higher rate of cerebrovascular events, serious hemorrhage complications, and all-cause death in the elderly group with AF ablation therapy than the younger group, but these results might be affected by potentially inherent confounding factors. The elderly might be associated with more frail profiles, higher stroke/bleeding risk, and a higher rate of comorbidities than younger individuals [3], which would overestimate our results owing to forward-causality bias. However, the pooled rates of safety outcomes in the elderly group were remarkably low, ranging from 0.00 to 0.01. Finally, the only potentially significant treatment–covariate interaction was identified in the follow-up time subgroup: ≥24 months and <24 months with *p* = 0.066 for interaction for the comparison of freedom rate from AF between elderly and younger groups. Meanwhile, important confounding factors, including LAD and AF history duration, showed no effect on the rate of freedom from AF ablation between elderly and younger groups, whereas the relatively few eligible studies are reason to interpret these results with more caution. Therefore, more studies with larger samples and longer follow-up are needed to confirm our results.

## 6. Conclusions

Our meta-analysis suggests that elderly patients may have inferior efficacy and safety outcomes to younger patients with AF ablation. Moreover, the follow-up time may be a potential determinant of outcome impact on freedom from AF for elderly patients after AF ablation. Additional randomized controlled trials are required for confirmation of our results.

## Figures and Tables

**Figure 1 jcm-11-04468-f001:**
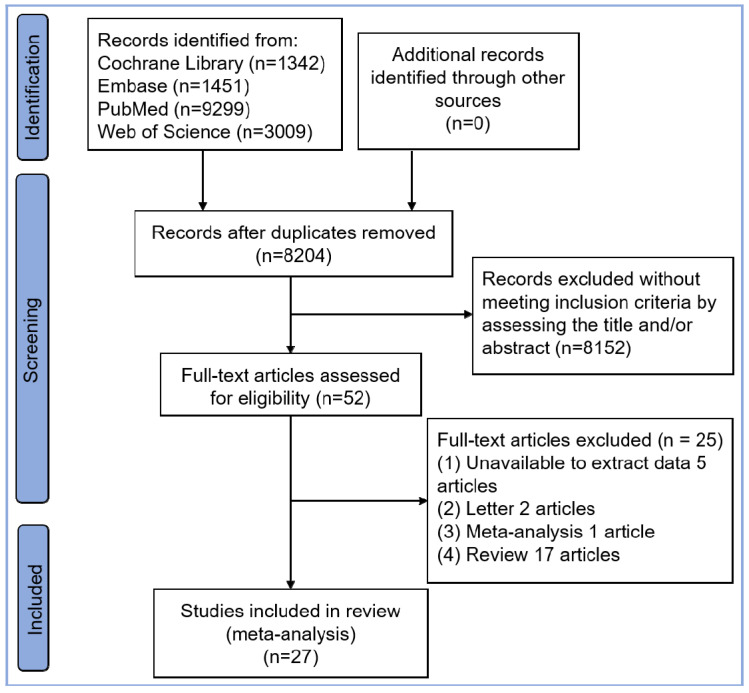
Flowchart of study selection.

**Figure 2 jcm-11-04468-f002:**
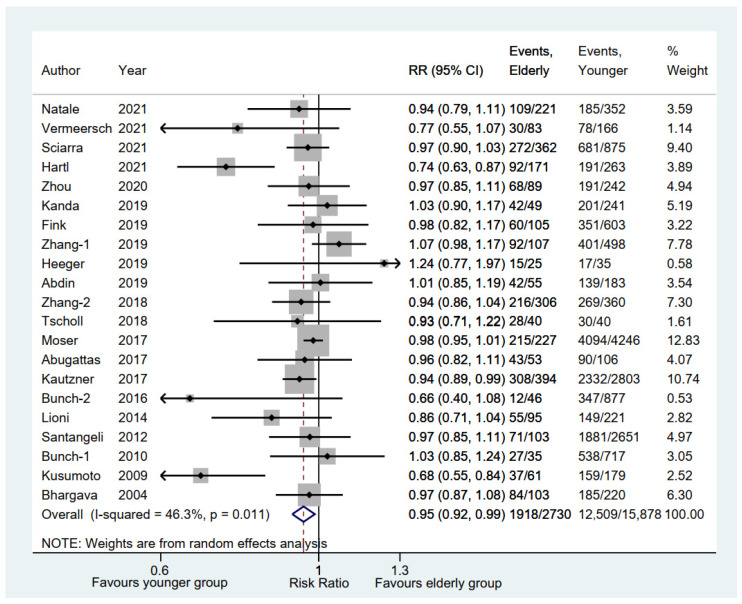
Forest plot of the freedom rate from AF between elderly and younger groups [6,10,11,12,13,14,15,16,17,19,20,21,22,23,24,25,26,27,29,30,31]. Comparison of the rates of freedom from AF between elderly and younger groups.

**Figure 3 jcm-11-04468-f003:**
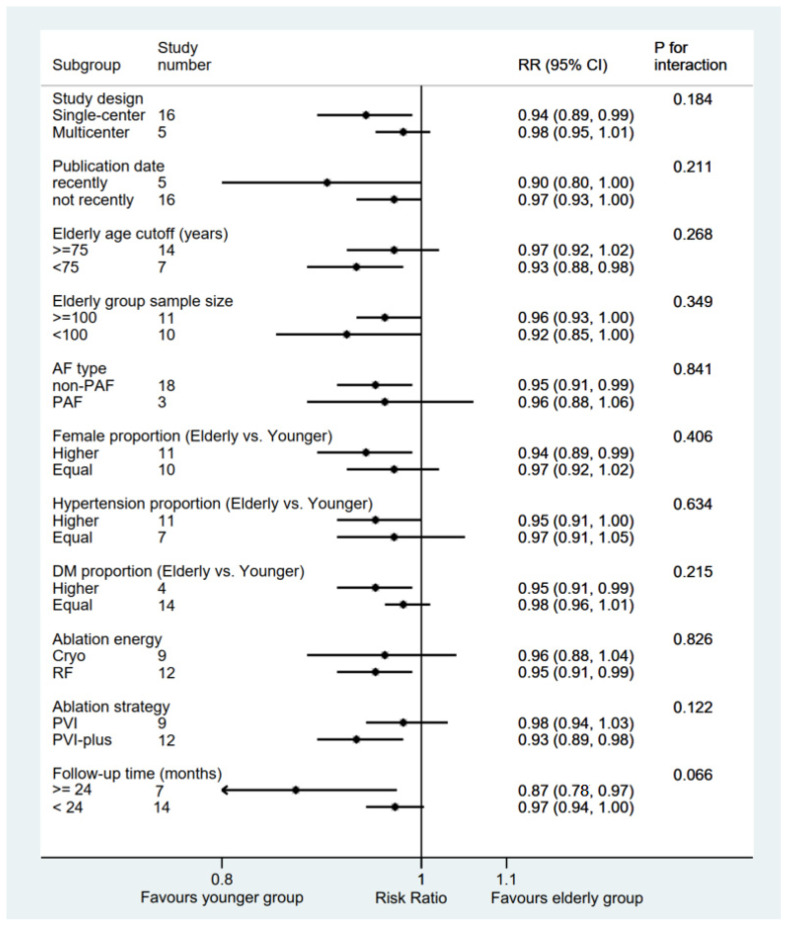
Forest plot of subgroup analysis of the freedom rate from AF between elderly and younger groups. Subgroup analysis of the rates of freedom from AF between elderly and younger groups. AF: atrial fibrillation; PAF: paroxysmal atrial fibrillation; RF: radiofrequency; Cryo: cryoablation; PVI: pulmonary vein isolation; PVI-plus: PVI plus linear and/or substrate ablation.

**Figure 4 jcm-11-04468-f004:**
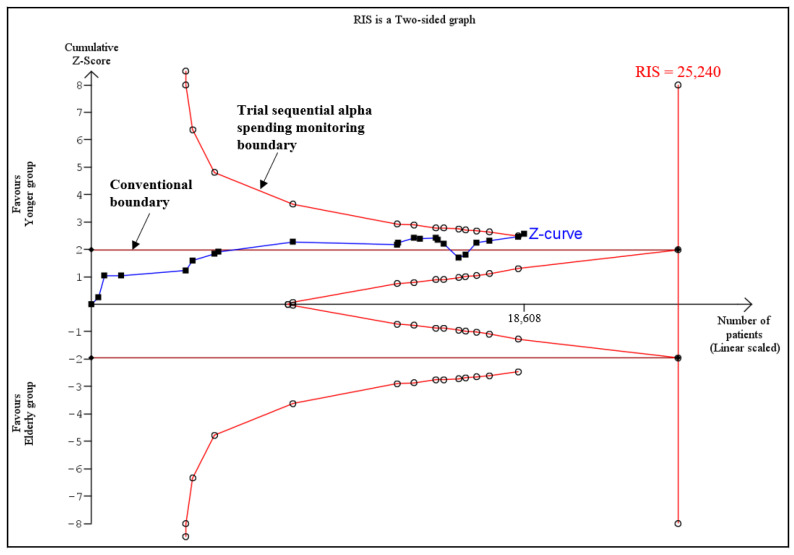
Trial sequential analysis of the rates of freedom from AF between elderly and younger groups. The results showed that the actual sample size (18,608) was smaller than the RIS (relative risk reduction, RRR = 35%; RIS = 25,240), and the cumulative Z curve (Z = 2.55) crossed both the conventional boundary and the trial sequential alpha spending monitoring boundary (TSA monitoring boundaries = 2.49). RIS: required information size.

**Figure 5 jcm-11-04468-f005:**
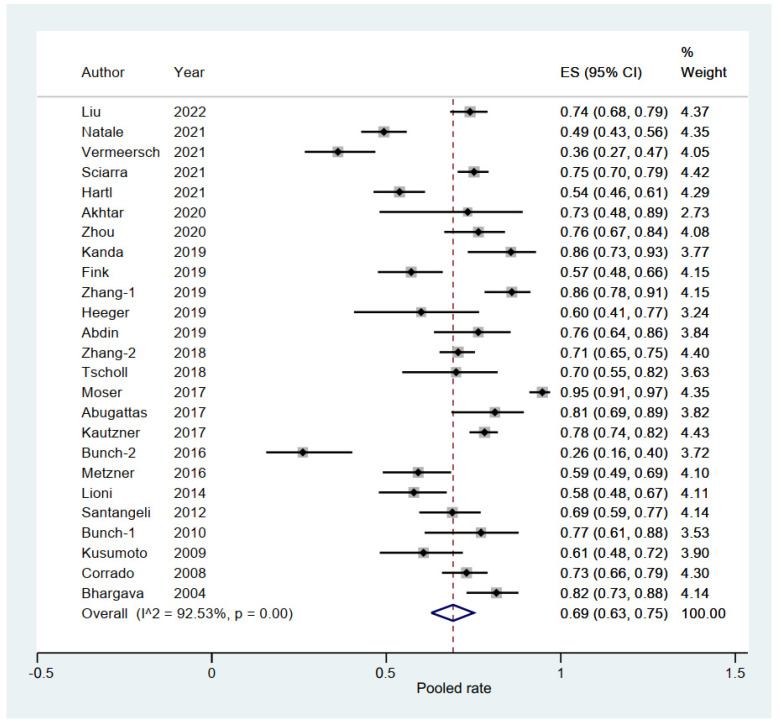
Forest plot of the pooled rate of freedom from AF in the elderly group [6,10,11,12,13,14,15,16,17,19,20,21,22,23,24,25,26,27,29,30,31,32,33,34,35]. The line of equity refers to the pooled result of eligible studies in the forest plots.

**Figure 6 jcm-11-04468-f006:**
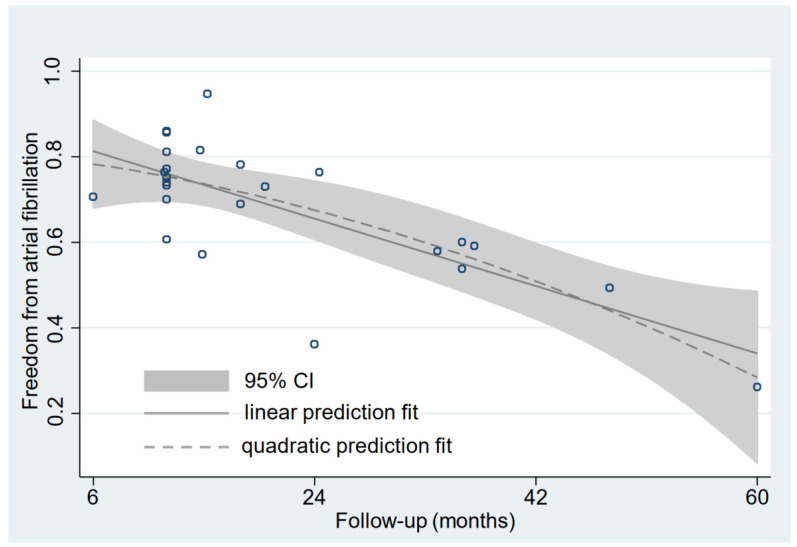
Linear and quadratic prediction fit plot with confidence interval between the follow-up time and the rate of freedom from AF for the elderly. The solid gray line represents the linear prediction fit. The dotted gray line represents the quadratic prediction fit. The gray area represents the 95% confidence interval for the quadratic prediction fit. Each dark blue circle represents a sample. CI: confidence interval.

**Figure 7 jcm-11-04468-f007:**
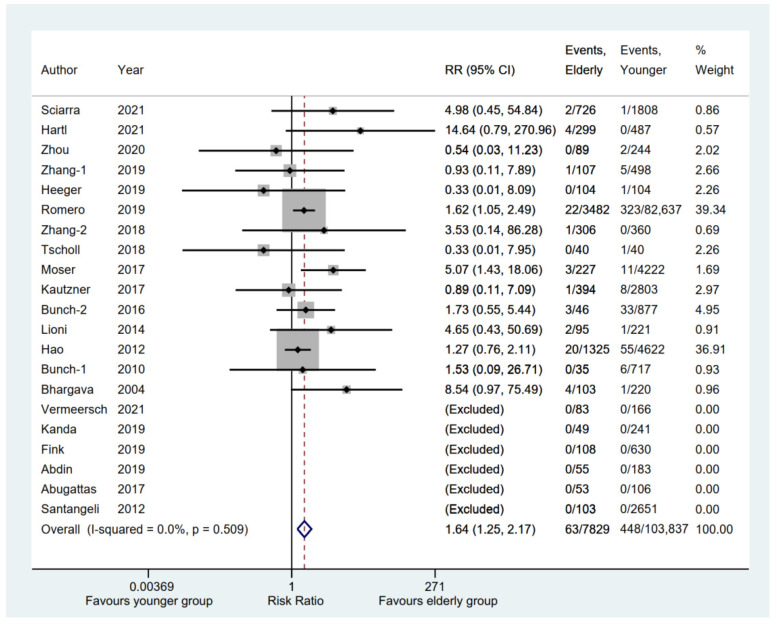
Forest plot comparing cerebrovascular events between elderly and younger groups [6,11,12,13,14,15,16,17,18,19,20,21,22,23,24,25,26,27,28,29,31]. The dotted red line represents the pooled risk ratio for cerebrovascular events between elderly and younger groups.

**Figure 8 jcm-11-04468-f008:**
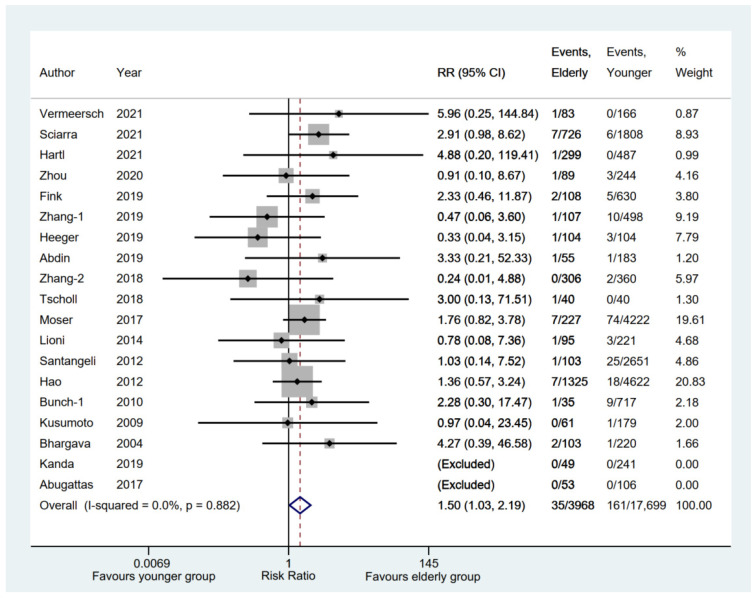
Forest plot comparing serious hemorrhage complications between elderly and younger groups [6,11,12,13,14,15,16,17,19,20,21,22,23,26,27,28,29,30,31]. The dotted red line represents the pooled risk ratio for serious hemorrhage complications between elderly and younger groups.

**Figure 9 jcm-11-04468-f009:**
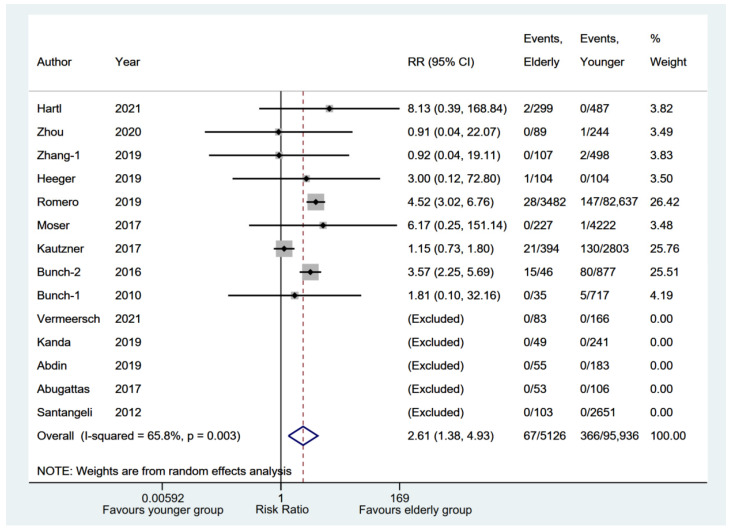
Forest plot comparing all-cause death between elderly and younger groups [6,12,13,14,16,17,18,19,22,23,24,25,27,29]. The dotted red line represents the pooled risk ratio for all-cause death between elderly and younger groups.

**Figure 10 jcm-11-04468-f010:**
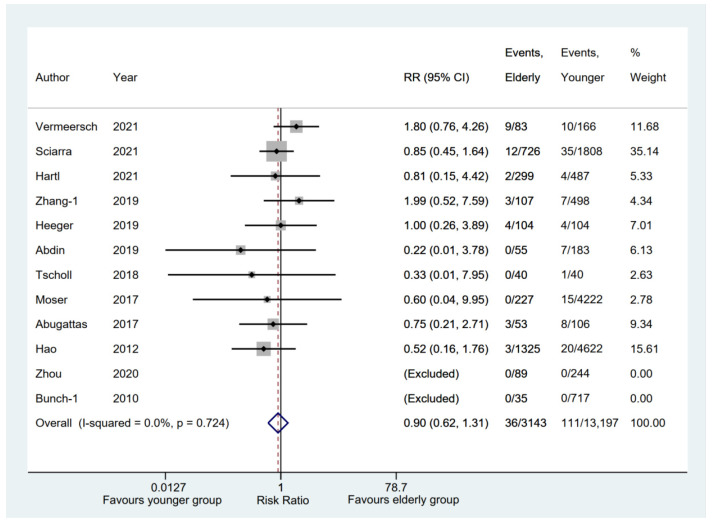
Forest plot comparing phrenic nerve injury between elderly and younger groups [6,11,12,13,16,17,19,21,22,23,28,29]. The dotted red line represents the pooled risk ratio for phrenic nerve injury between elderly and younger groups.

**Table 1 jcm-11-04468-t001:** Baseline characteristics and procedure-related indices of eligible studies.

**First** **Author**	**Year**	**Study Design**	**Country**	**Sample Size**		**Elderly Age Cutoff (Years)**	**Gender (Female, %)**		**AF Type (PAF, %)**		**Hypertension (%)**		**DM (%)**	
				**Elderly Group**	**Younger Group**		**Elderly Group**	**Younger Group**	**Elderly Group**	**Younger Group**	**Elderly Group**	**Younger Group**	**Elderly Group**	**Younger Group**
Natale [10]	2021	Observational single-center	America	221	352	75	100.0	100.0	10.9	14.8	69.7 ^$^	53.4	10.9	14.8
Vermeersch [6]	2021	Retrospective single-center	Belgium	83	166	75	41.0	39.2	0	0	73.5 ^$^	55.4	14.5	13.9
Sciarra [11]	2021	Prospective multicenter	Italy	726	1808	67	37.5 ^#^	23.4	73.6	75.5	62.8 ^$^	41.3	7.2 ^&^	4.7
Hartl [12]	2021	Observational single-center	Germany	299	487	70	44.5 ^#^	41.9	49.2	70.0	73.6 ^$^	60.0	NA	NA
Zhou [13]	2020	Observational single-center	China	89	244	80	55.1	59.4	64.0	62.7	75.3	64.8	36.0 ^&^	22.1
Kanda [14]	2019	Retrospective single-center	Japan	49	241	80	51.0	40	100.0	100.0	63.0	56.0	14.0	16.0
Fink [15]	2019	Prospective multicenter	Germany	108	630	70	38.0 ^#^	27.6	55.6	63.0	NA	NA	9.3	6.3
Zhang-1 [16]	2019	Retrospective single-center	China	127	550	75	55.1 ^#^	40.5	92.9	88.2	67.2 ^$^	59.8	18.1	15.1
Heeger [17]	2019	Prospective multicenter	Germany	104	104	75	50.0	48.1	57.7	56.7	77.9	78.9	15.4	14.4
Romero [18]	2019	Retrospective multicenter	America	3482	82,637	80	61.0	31.1	NA	NA	65.7	55.3	19.6	14.9
Abdin [19]	2019	Retrospective single-center	Germany	55	183	75	54.6 ^#^	34.5	31.0	40.5	85.4 ^$^	69.3	20.0 ^&^	8.7
Zhang-2 [20]	2018	Retrospective single-center	China	308	360	60	41.5 ^#^	21.4	71.4	75.0	55.0 ^$^	37.0	12.6	9.9
Tscholl [21]	2018	Retrospective single-center	Germany	40	40	75	50.0	35	45.0	47.5	80.0	60.0	10.0	12.5
Moser [22]	2017	Retrospective multicenter	Germany	227	4222	75	48.0 ^#^	31.1	59.9	63.3	NA	NA	8.8	7.6
Abugattas [23]	2017	Retrospective single-center	Belgium	53	106	75	54.7	41.5	100.0	100.0	79.2 ^$^	41.3	11.3	8.7
Kautzner [24]	2017	Retrospective single-center	Czech Republic	394	2803	70	49.0 ^#^	29.4	66.5	68.2	79.2 ^$^	56.7	15.7 ^&^	11.2
Bunch-2 [25]	2016	Observational multicenter	America	46	877	80	58.7 ^#^	40.2	52.2	54.7	82.6	70.0	17.4	22.1
Lioni [26]	2014	Retrospective single-center	Greece	95	221	65	49.5	41.2	100.0	100.0	41.1	33.5	20.0 ^&^	6.8
Santangeli [27]	2012	Retrospective single-center	America	103	2651	80	41.0 ^#^	28.0	25.0	27.0	48.0 ^$^	37.0	15.0	11.0
Hao [28]	2012	Retrospective multicenter	America	1325	4622	65	41.0 ^#^	23.0	NA	NA	68.0	57.0	21.0	16.0
Bunch-1 [29]	2010	Retrospective single-center	America	35	717	80	54.3	40.7	45.7	54.1	57.1	49.2	8.6	12.4
Kusumoto [30]	2009	Retrospective single-center	America	61	179	75	39.3 ^#^	24.1	34.0	70.9	NA	NA	NA	NA
Bhargava [31]	2004	Retrospective single-center	America	103	220	60	23.3	18.2	52.4	54.5	35.0 ^$^	20.9	NA	NA
Liu [32]	2022	Multicenter single-arm	China	270	-	80	42.6	-	65.6	-	73.7	-	29.3	-
Akhtar [33]	2020	Single-center single-arm	America	15	-	80	40.0	-	87.0	-	80.0	-	20.0	-
Metzner [34]	2016	Single-center single-arm	Germany	94	-	75	41.5	-	58.5	-	88.3	-	4.3	-
Corrado [35]	2008	Single-center single-arm	America	174	-	75	36.8	-	55.0	-	56.0	-	13.0	-
**First** **Author**	**LVEF**		**CHA_2_DS_2_-VASc Score**		**LAD (mm)**			**AF History Duration**			**AADs Usage (Elderly vs. Younger)**			
	**Elderly Group**	**Younger Group**	**Elderly Group**	**Younger Group**	**Elderly Group**		**Younger Group**	**Elderly Group**		**Younger Group**				
Natale [10]	58.2 ± 9.6	57.8 ± 9.4	NA	NA	42.6 ± 7.8		41.9 ± 7.6	NA		NA	NA			
Vermeersch [6]	53.2 ± 9.4	54.4 ± 9.0	NA	NA	45.8 ± 7.8		45.6 ± 7.0	45.7 ± 46.2 M		52.4 ± 61.1 M	NA			
Sciarra [11]	58.8 ± 7.2	59.3 ± 6.9	2.4 ± 0.7	1.1 ± 0.9	22.9 ± 6.2 cm^2^ *		21.8 ± 6.0 cm^2^	62.0 ± 107.1 M ^ξ^		52.0 ± 105.8 M	Failed ≥2 AADs (higher)			
Hartl [12]	56.2 ± 5.9	56.8 ± 6.5	NA	NA	46.1 ± 7.0 *		43.5 ± 6.9	NA		NA	The proportion of AADs at baseline (equal)			
Zhou [13]	62.7 ± 5.4	63.1 ± 5.7	4.3 ± 1.3	3.3 ± 1.4	41.2 ± 4.8		41.5 ± 6.2	12.0 (2.5-36.0) M		24.0 (5.0-48.0) M	NA			
Kanda [14]	NA	NA	3.8 ± 0.9	2.2 ± 1.4	40.0 ± 6.0		38.0 ± 6.0	NA		NA	The proportion of AADs at baseline: Class I (lower), other classes (equal)			
Fink [15]	NA	NA	NA	NA	NA		NA	NA		NA	NA			
Zhang-1 [16]	58.7 ± 9.0	61.5 ± 6.5	4.8 ± 1.6	2.6 ± 1.7	41.0 ± 5.3		41.3 ± 5.6	NA		NA	The proportion of AADs at baseline, Class I, I and III (equal)			
Heeger [17]	NA	NA	3.8 ± 1.1	2.1 ± 1.3	44.5 ± 5.6		44.5 ± 5.6	NA		NA	NA			
Romero [18]	NA	NA	NA	NA	40.8 ± 5.5		40.8 ± 6.6	24.6 ± 34.1 M		21.9 ± 34.6 M	NA			
Abdin [19]	51.6 ± 8.3	52.5 ± 8.0	4.0 ± 1.3	2.0 ± 1.3	49.2 ± 5.8		38.6 ± 6.1	NA		NA	NA			
Zhang-2 [20]	66.3 ± 5.7	69.1 ± 8.9	NA	NA	NA		NA	NA		NA	NA			
Tscholl [21]	63.0 (60.0, 66.0)	65.0 (60.0, 70.0)	4.0 (4.0, 5.0)	2.0 (1.0, 3.0)	NA		NA	NA		NA	NA			
Moser [22]	NA	NA	3.7 ± 1.0	1.7 ± 1.2	41.4 ± 7.2		40.9 ± 6.6	NA		NA	NA			
Abugattas [23]	59.2 ± 5.2	59.9 ± 6.4	4.0 ± 1.3	1.3 ± 1.2	42.5 ± 5.4		42.3 ± 5.7	NA		NA	NA			
Kautzner [24]	55.8 ± 8.8	56.4 ± 7.6	3.1 ± 1.3	1.5 ± 1.2	NA		NA	NA		NA	The proportion of AADs at baseline (equal)			
Bunch-2 [25]	53.8 ± 13.3	52.5 ± 11.4	NA	NA	41.2 ± 4.8		41.5 ± 6.2	12.0 (2.5-36) M		24.0 (5.0-48.0) M	NA			
Lioni [26]	60.0 ± 3.8	61.1 ± 4.0	NA	NA	42.6 ± 4.5 *		39.5 ± 4.3	5.9 ± 5.1 Y ^ξ^		4.7 ± 4.4 Y	The proportion of AADs after ablation, Class I and III (equal)			
Santangeli [27]	55.0 ± 12.0	57.0 ± 9.0	NA	NA	46.0 ± 5.0		45.0 ± 8.0	52.0 (24.0-78.0) M		58.0 (31.0-96.0) M	Failed AADs (equal)			
Hao [28]	NA	NA	NA	NA	24.8 ± 9.1 cm^2^		28.7 ± 9.5 cm^2^	NA		NA	NA			
Bunch-1 [29]	52.7 ± 13.2	51.3 ± 13.1	NA	NA	NA		NA	NA		NA	NA			
Kusumoto [30]	NA	NA	NA	NA	42.6 ± 4.5		39.1 ± 4.3	5.9 ± 5.1 Y		4.7 ± 4.4 Y	The proportion of AADs after ablation, Class I and III (equal)			
Bhargava [31]	51.4 ± 9.8	53.4 ± 7.6	NA	NA	43.4 ± 6.5		43.3 ± 13.2	6.5 ± 3.7 Y		6.0 ± 4.8 Y	Failed AADs (equal)			
Liu [32]	63.7 ± 7.2	-	3.9 ± 1.2	-	39.9 ± 6.3		-	2.9 ± 5.2 Y		-	-			
Akhtar [33]	63.7 ± 3.5	-	4.2 ± 1.7	-	45.0 ± 1.2		-	8.9 ± 8.2 Y		-	-			
Metzner [34]	NA	-	4.0 ± 1.0	-	44.8 ± 6.2		-	75.0 M Median		-	-			
Corrado [35]	53.0 ± 7.0	-	NA	-	46.0 ± 6.0		-	7.0 ± 4.0 Y		-	-			
**First Author**	**Key Points of Ablation Procedure**							**Ablation Strategy**		**Ablation Energy**	**Follow-Up (Months)**			
Natale [10]	Isolation of pulmonary veins, posterior wall and superior vena cava was performed in all patients. Non-pulmonary vein triggers from other areas were ablated based on operator’s discretion							PVI-plus		RF	48.0			
Vermeersch [6]	PVI only							PVI		Cryo	24.0 (18.4-25.5)			
Sciarra [11]	PVI only							PVI		Cryo	12.0			
Hartl [12]	PVI with or without additional linear ablation based on decision							PVI-plus		Cryo	36.0			
Zhou [13]	After PVI, additional linear ablation was performed when necessary							PVI-plus		RF	24.4 ± 9.6			
Kanda [14]	PVI with or without additional linear ablation based on decision							PVI-plus		Cryo	12.0			
Fink [15]	PVI first, and then additional ablation strategies including the creation of right atrial and left atrial linear lesions including block of the cavo-tricuspid isthmus, or ablation of complex fractionated atrial electrograms were at the discretion of the operator							PVI-plus		RF	14.9			
Zhang-1 [16]	PVI only							PVI		Cryo	12.0			
Heeger [17]	PVI only							PVI		Cryo	36.0			
Romero [18]	NA							NA		NA	NA			
Abdin [19]	PVI only							PVI		Cryo	11.8 ± 5.4			
Zhang-2 [20]	PVI with linear ablation							PVI-plus		RF	6.0			
Tscholl [21]	PVI only							PVI		Cryo	12.0 (6.0, 18.0)			
Moser [22]	PVI first, and then ablation of fragmented signals and/or lines in the left atrial (mitral isthmus line, roof line, anterior line) were performed in order to achieve termination to sinus rhythm							PVI-plus		RF	15.3			
Abugattas [23]	PVI only							PVI		Cryo	12.0			
Kautzner [24]	All patients underwent PVI first, and then additional left atrial linear lesions, coronary sinus ablation, or electrogram-guided ablations were performed empirically according to the clinical presentation and inducibility of the arrhythmia during the procedure							PVI-plus		RF	18.0-21.0			
Bunch-2 [25]	All patients underwent PVI first, and then additional ablation beyond PVI was performed based upon individual operator choice							PVI-plus		RF	60.0			
Lioni [26]	PVI only							PVI		RF	34.0 ± 15.1			
Santangeli [27]	Isolation of all the pulmonary vein antra and the posterior wall contained between the pulmonary veins first; then the ablation catheter was positioned at right atrium-superior vena cava junction, where mapping and ablation was performed.							PVI-plus		RF	18.0 ± 6.0			
Hao [28]	NA							NA		RF	1.0 W			
Bunch-1 [29]	PVI with or without additional linear ablation based on decision							PVI-plus		RF	12.0			
Kusumoto [30]	PVI with linear ablations (not routinely performed)							PVI-plus		RF	12.0			
Bhargava [31]	PVI only							PVI		RF	14.7 ± 5.2			
Liu [32]	PVI with or without additional linear ablation based on decision							PVI-plus		RF	12.0			
Akhtar [33]	PVI first, then additional cavo-tricuspid isthmus ablation based on the discretion of the operator							PVI-plus		Cryo	12.0			
Metzner [34]	Circumferential PVI was performed in all patients, then ablation of complex fractionated atrial electrograms and/or linear lesions were performed based on decision							PVI-plus		RF	37.0 ± 20.0			
Corrado [35]	PVI and superior vena isolation							PVI-plus		RF	20.0 ± 14.0			

AF: atrial fibrillation; PAF: paroxysmal atrial fibrillation; DM: diabetes mellitus; LVEF: left ventricular ejection fraction; LAD: left atrial diameter; AADs: antiarrhythmic drugs; PVI: pulmonary vein isolation; PVI-plus: PVI plus linear ablation and/or substrate ablation RF: radiofrequency; Cryo: cryoablation; NA: not available. Note: ^#^, ^$^, ^&^, *, and ^ξ^ represent the significantly higher proportion (elderly group vs. younger group) in terms of gender, hypertension, DM, LAD, and AF history duration, respectively. In the LAD column, cm^2^ represents the unit of left atrial area; in the AF history duration column, M and Y represent months and years, respectively. In the Follow-up column, W represents week.

**Table 2 jcm-11-04468-t002:** Subgroup analysis of the rate of freedom from atrial fibrillation in the elderly group.

Subgroup Factors	Numbers in Study	Pooled Incidence	95% CI	I^2^ (%)	*p* for Interaction
Study design					0.893
Multicenter	7	0.68	0.54–0.81	95.74	
Single-center	18	0.69	0.63–0.76	89.38	
Publication date					0.189
recently	7	0.63	0.51–0.74	93.52	
not recently	18	0.72	0.65–0.78	91.20	
Elderly age cutoff (years)					0.911
≥75	18	0.69	0.60–0.78	93.54	
<75	7	0.69	0.61–0.76	89.64	
Elderly group sample size					0.264
≥100	13	0.72	0.65–0.79	94.17	
<100	12	0.65	0.55–0.75	87.76	
AF type					0.484
PAF	3	0.75	0.56-0.91	-	
non-PAF	22	0.68	0.62–0.75	93.12	
Ablation strategy					0.771
PVI-plus	16	0.68	0.60–0.76	93.85	
PVI	9	0.70	0.60–0.80	89.60	
Ablation energy					0.765
RF	15	0.68	0.60–0.76	93.88	
Cryo	10	0.71	0.60–0.80	90.15	
Follow-up time (months)					0.000
≥24	8	0.53	0.43–0.62	85.67	
<24	17	0.76	0.71–0.81	86.11	

AF: atrial fibrillation; PAF: paroxysmal atrial fibrillation; PVI: pulmonary vein isolation; PVI plus: PVI plus linear and/or complex fractionated atrial electrogram ablations; CI: confidence interval.

## Data Availability

The data that support the findings of this study are available from the corresponding author upon reasonable request.

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
