# Peer review of "Do Elderly Patients with Atrial Fibrillation Have Comparable Ablation Outcomes Compared to Younger Ones? Evidence from Pooled Clinical Studies"

_jcm, 2022, doi:10.3390/jcm11154468_

Round 1

Reviewer 1 Report

This paper evaluated the ablation outcomes between elderly and younger patients with AF using a meta-analysis. Compared with the younger group, the elderly group was significantly associated with a lower rate of freedom from AF, as well as a higher incidence of safety outcomes. The study suggests that elderly patients may have inferior efficacy and safety outcomes to younger patients with AF ablation.

This paper is dealing with an important topic. 

  1. Please cite recent publications about catheter ablation and rhythm control in the elderly population (PMID: 35589174, PMID: 33731545).
  2. Can you present the effect of ablation according to age?
  3. Cryoablation is relatively safer than catheter ablation. Can you analyze the effect of cryoablation in elderly AF patients in a subgroup analysis?

Author Response

Point 1: Please cite recent publications about catheter ablation and rhythm control in the elderly population (PMID: 35589174, PMID: 33731545).

Response 1: Thank you for your helpful suggestion. We have cited these publications as the reference 4 and 5, which would facilitate improve our manuscript.

Point 2: Can you present the effect of ablation according to age?

Response 2: A good question. The effects of rhythm control (e.g., antiarrhythmic drugs or ablation) according to age is an interesting and promising topic. Similarly, our study aims to explore the ablation outcomes (including efficacy and safety outcomes) between elderly and younger patients with AF. The individual data (e.g., age, ablation outcomes) in each eligible study of our meta-analysis is unavailable, although we have tried to contact the corresponding authors for the original data in their publications. Therefore, this study aims to assess the effect of ablation according to two groups (elderly group and younger group, defined by the elderly age cutoff value), not to individual age. Still, our result suggests that elderly patients may have inferior efficacy to younger patients with AF ablation, which is consistent with the recent results from a real-world analysis (DOI: 10.1253/circj.CJ-20-1062).

Point 3: Cryoablation is relatively safer than catheter ablation. Can you analyze the effect of cryoablation in elderly AF patients in a subgroup analysis?

Response 3: Thank you for your constructive question. As your mentioned, cryoablation is relatively safer than catheter ablation and has become an initial therapy for atrial fibrillation (PMID: 34042402). However, the effect of cryoablation in elderly AF patients is still controversial. Compared with younger patients, some studies showed inferior results in elderly patients (PMID: 19148720; PMID: 34141015), while others showed comparable results (PMID: 31507014; PMID: 30287056). Interestingly, subgroup analysis in our meta-analysis revealed that elderly group has a similar effect to the younger group in the cryoablation subgroup (RR=0.96; 95%CI: 0.88-1.04, P>0.05; Figure 3). Although the relevant mechanism is unclear, the subgroup-analysis results are expected to play a potential guiding role in the optimal management of AF for elderly individuals. Therefore, we highlight these results are still required to be demonstrated with more randomized controlled trials (in Discussion section; line 414-415)

###For additional revisions###

  1. In Results section “3.5. The safety outcomes between elderly and younger groups”, figures for cerebrovascular events, serious hemorrhage complications, and phrenic nerve injury are replaced by new figures with a fixed effect model, and the corresponding data also be updated without changing our previous results.
  2. A note is added under Table 1. “Note: #, $, and & represent the significantly higher proportion (elderly group versus younger group) in terms of gender, hypertension, and DM, respectively.” (line 166-168)
  3. In addition to correct this manuscript according to your advice, we also checked the manuscript time and time again to avoid grammatical or spelling errors throughout the article. Hope our corrected manuscript can give you satisfaction. That will be our great honor.

Reviewer 2 Report

In this systematic review associated with meta-analysis, authors tried to compare the outcomes of AF ablation among elderly and younger patients. They found that elderly group had a significantly lower rate of freedom from AF (risk ratio [RR] 0.95) in comparison with younger group. Moreover, a higher incidence of major adverse outcomes (cerebrovascular events: RR, 1.64; serious hemorrhage complications: RR, 1.60; all-cause death: RR, 2.61). Based on the subgroup analysis (baseline demographics and interventional features) and quadratic prediction fit analysis, the follow-up duration (≥24 months vs. <24 months) was the potential determinant on freedom from AF for elderly patients. In this comprehensive systematic review and meta-analysis, the study flow is appropriate for methods and analyses/interpretations, but some points need to be mentioned here for improving the study presentation:

1) Regarding subgroup analysis, some factors can significantly affect the rate of freedom from AF ablation, including drug consumption before and after ablation procedure, the presence of structural heart diseases, AF duration, and left atrial diameter. Why have not authors implemented these important parameters for subgroup analysis? The relationship between time and the rate of freedom from AF is inevitable. We need to explore more important confounding factors acting as the main predictors of AF recurrence after AF ablation procedure.  

2) Analyses for outcomes among only elderly patients without comparing with younger patients can be omitted from the main text and transferred to supplementary file, because the main aim of meta-analysis is for comparing outcomes of AF ablation between elderly and younger population.

3) Tables 2 and 3 can be reported as supplementary files. Instead, Suppl. Figure 1 (TSA) can be reported as a main figure within the text

Author Response

Point 1: Regarding subgroup analysis, some factors can significantly affect the rate of freedom from AF ablation, including drug consumption before and after ablation procedure, the presence of structural heart diseases, AF duration, and left atrial diameter. Why have not authors implemented these important parameters for subgroup analysis? The relationship between time and the rate of freedom from AF is inevitable. We need to explore more important confounding factors acting as the main predictors of AF recurrence after AF ablation procedure.

Response 1: Thank you for your constructive suggestions. It is an important and permanent topic for clinical EP doctors to explore the clinical confounding factors acting as the main predictors of AF recurrence after AF ablation procedure. As your mentioned, multiple clinical characteristics (e.g., drug consumption, structural heart diseases, AF duration, LAD) can significantly affect the rate of freedom from AF ablation. Whereas, the relevant data (e.g., AF duration, LAD, and drug consumption) in some eligible studies of our meta-analysis is not available, though we have tried to contact the corresponding authors for the data in their publications. Still, other clinical key factors (e.g., AF type, follow-up time, ablation strategy, ablation energy, comorbidities, and so on) are analyzed in our subgroup analysis, which has been discussed in the Discussion section. As your mentioned, the relationship between time and the rate of freedom from AF is inevitable. Whereas, multiple studies revealed the overall success rate (freedom from AF) was comparable between elderly group and younger group with various follow-up time (PMID: 32357382; PMID: 31507014; PMID: 30287056), while some recent studies show the different results (PMID: 34141015; PMID: 19148720). Meanwhile, no classes of recommendations to date are assigned in AF guidelines. The above indicates a potentially unresolved controversy on the AF ablation therapy for the elderly. By pooling the clinical studies, we found elderly patients may have inferior efficacy and safety outcomes to younger patients with AF ablation, and subgroup analysis and quadratic prediction fit analysis revealed the follow-up time might be the potential determinant on freedom from AF for elderly patients after AF ablation. Our results might provide potential guidance for AF ablation for the elderly.

Point 2: Analyses for outcomes among only elderly patients without comparing with younger patients can be omitted from the main text and transferred to supplementary file, because the main aim of meta-analysis is for comparing outcomes of AF ablation between elderly and younger population.

Response 2: We sincerely appreciate your careful review. We performed this meta-analysis to compare the ablation outcomes between elderly and younger patients with AF and identify possible determinants of outcome impact for elderly patients after ablation. In the Results section, we reported the rate of freedom from AF between elderly and younger group, and subgroup analysis showed the only potentially detrimental factor was follow-up time. For further analysis and explanation of the subgroup analysis, two key results (the “3.3. The pooled rate of freedom from AF in elderly group” and “3.4. The relationship between the follow-up time and the rate of freedom from AF”) are presented in the main text, which also contributes to develop the second main finding (line 331-332).

Point 3: Tables 2 and 3 can be reported as supplementary files. Instead, Suppl. Figure 1 (TSA) can be reported as a main figure within the text

Response 3: Thank you for the helpful suggestion. We have revised Table 2-3 and Supplementary Figure 1(TSA) based on your suggestions. “Table 2-3” changes to “Supplementary Table 1-2”, and “Supplementary Figure 1(TSA)” changes to “Figure 4”.

###For additional revisions###

  1. In Results section “3.5. The safety outcomes between elderly and younger groups”, figures for cerebrovascular events, serious hemorrhage complications, and phrenic nerve injury are replaced by new figures with a fixed effect model, and the corresponding data also be updated without changing our previous results.
  2. A note is added under Table 1. “Note: #, $, and & represent the significantly higher proportion (elderly group versus younger group) in terms of gender, hypertension, and DM, respectively.” (line 166-168)
  3. In addition to correct this manuscript according to your advice, we also checked the manuscript time and time again to avoid grammatical or spelling errors throughout the article. Hope our corrected manuscript can give you satisfaction. That will be our great honor.

Reviewer 3 Report

This paper has shown that the older patients with atrial fibrillation have higher risk of recurrence of ablation and merit of ablation for younger patients.

However, it is well known fact.

You should have shown the additional merit for younger and older patients for ablation.

You should have shown any data sch as echocardiography, biochemical markers.

Author Response

Point 1: This paper has shown that the older patients with atrial fibrillation have higher risk of recurrence of ablation and merit of ablation for younger patients. However, it is well known fact.

Response 1: We sincerely appreciate your careful review. Although the latest AF guidelines emphasized catheter ablation for selected elderly AF patients might be a safe and effective option with comparable success rates and acceptable complication incidence to younger AF patients, no classes of recommendations to date are assigned in AF guidelines. Moreover, compared with younger patients, some studies showed inferior results in elderly patients (PMID: 19148720; PMID: 34141015), while some studies showed comparable results (PMID: 31507014; PMID: 30287056). These all indicate a potentially unresolved controversy on the AF ablation therapy for the elderly. Therefore, this meta-analysis for evaluating the ablation outcome between elderly and younger patients with AF is necessary.

Point 2: You should have shown the additional merit for younger and older patients for ablation.

Response 2: Thank you for the helpful suggestion. Although our meta-analysis suggests that elderly patients may have inferior efficacy and safety outcomes to younger patients with AF ablation, the potential merits for ablation also are shown in Results section 3.3 (the pooled rate of freedom from AF in elderly group is still relatively high) and Results section 3.5 (the safety outcomes in elderly group are still relatively low). Moreover, we also highlight the potential reasons for the higher safety outcomes in elderly group vs younger group (shown in Limitation section, line 426-433).

Point 3: You should have shown any data sch as echocardiography, biochemical markers.

Response 3: Thank you for your constructive suggestions. Cochrane Library, Embase, PubMed, and Web of Science were systematically searched in our study. Whereas the relevant data (e.g., AF duration, LAD, and drug consumption, some biochemical markers) in some eligible studies of our meta-analysis is not available, though we have tried to contact the relevant corresponding authors for the data in their publications. Consistent with your helpful suggestion, we have identified and presented multiple key information and characteristics (including study design, sample size, gender, AF type, hypertension, DM, LVEF, CHA2DS2-VASc score, ablation strategy, ablation energy, and follow-up time), which would facilitate a systematic and comprehensive presentation of each eligible study (shown in Table 1).

###For additional revisions###

  1. In Results section “3.5. The safety outcomes between elderly and younger groups”, figures for cerebrovascular events, serious hemorrhage complications, and phrenic nerve injury are replaced by new figures with a fixed effect model, and the corresponding data also be updated without changing our previous results.
  2. A note is added under Table 1. “Note: #, $, and & represent the significantly higher proportion (elderly group versus younger group) in terms of gender, hypertension, and DM, respectively.” (line 166-168)
  3. In addition to correct this manuscript according to your advice, we also checked the manuscript time and time again to avoid grammatical or spelling errors throughout the article. Hope our corrected manuscript can give you satisfaction. That will be our great honor.

Round 2

Reviewer 2 Report

Thank to authors for their efforts in preparing the revised version of the manuscript. Unfortunately the main limitation of study has not been addressed in this revision, since the other major confounding factors (drug history and AF duration and LAD) have not been searched well within included studies.   

Author Response

Point 1: Thank to authors for their efforts in preparing the revised version of the manuscript. Unfortunately, the main limitation of study has not been addressed in this revision, since the other major confounding factors (drug history and AF duration and LAD) have not been searched well within included studies.   

Response 1: Thank you for your constructive suggestions, which would facilitate significantly improve our manuscript. According to your advice, we systematically searched all the eligible studies on the other major confounding factors, including LAD, AF history duration and AADs usage, which were added in Table 1. Moreover, subgroup analysis was performed to explore these additional confounding factors for the rate of freedom from AF based on the ‘equal’ and ‘higher’ subgroups (2.5. Statistical analysis; line 134-136), and the results were presented in the Supplementary Figure 1 (line 189-190). We also analyzed these results in the Discussion section (line 425-431) and Limitation section (line 468-471).

Reviewer 3 Report

This paper has shown that the older patients with atrial fibrillation have higher risk of recurrence of ablation and merit of ablation for younger patients.

You searched the difference between younger and older patients.

However, it is well known fact.

You should have shown the additional merit for younger and older patients for ablation.

You should have shown any data such as echocardiography, biochemical markers.

Author Response

Point 1: However, it is well known fact.

Response 1: We sincerely appreciate your careful review. Although the latest AF guidelines emphasized catheter ablation for selected elderly AF patients might be a safe and effective option with comparable success rates and acceptable complication incidence to younger AF patients, no classes of recommendations to date are assigned in AF guidelines. Moreover, compared with younger patients, some studies showed inferior results in elderly patients (PMID: 34141015; PMID: 19148720), while some studies showed comparable results (PMID: 30287056; PMID: 31507014). These all indicate a potentially unresolved controversy on the AF ablation therapy for the elderly. Therefore, this meta-analysis for evaluating the ablation outcome between elderly and younger patients with AF may be necessary.

Point 2: You should have shown the additional merit for younger and older patients for ablation.

Response 2: Thank you for the helpful suggestion. Although our meta-analysis suggests that elderly patients may have inferior efficacy and safety outcomes to younger patients with AF ablation, the potential merits for ablation also are shown in Results section 3.3 (the pooled rate of freedom from AF in elderly group is still relatively high) and Results section 3.5 (the safety outcomes in elderly group are still relatively low). Moreover, we also highlight the potential reasons for the higher safety outcomes in elderly group vs. younger group (shown in Limitation section, line 457-464).

Point 3: You should have shown any data such as echocardiography, biochemical markers.

Response 3: Thank you for your constructive suggestions. We systematically searched all the eligible studies on additional confounding factors, including LAD, AF history duration and AADs usage, which were added in Table 1. Moreover, subgroup analysis was performed to explore these additional confounding factors for the rate of freedom from AF based on the ‘equal’ and ‘higher’ subgroups (2.5. Statistical analysis; line 134-136), and the results were presented in the Supplementary Figure 1 (line 189-190). We also analyzed these results in the Discussion section (line 425-431) and Limitation section (line 468-471).